# Anti-Ischemic Effects of PIK3IP1 Are Mediated through Its Interactions with the ET_A_-PI3Kγ-AKT Axis

**DOI:** 10.3390/cells11142162

**Published:** 2022-07-11

**Authors:** Jei Hyoung Park, Kyoung Jin Nho, Ji Young Lee, Yung Joon Yoo, Woo Jin Park, Chunghee Cho, Do Han Kim

**Affiliations:** School of Life Sciences, Gwangju Institute of Science and Technology (GIST), 123 Cheomdangwagi-ro, Buk-gu, Gwangju 61005, Korea; pjh900807@gist.ac.kr (J.H.P.); nkj1130@gist.ac.kr (K.J.N.); ljy0213@gist.ac.kr (J.Y.L.); yjyoo@gist.ac.kr (Y.J.Y.); wjpark@gist.ac.kr (W.J.P.); choch@gist.ac.kr (C.C.)

**Keywords:** myocardial infarction, ROS, PI3Kγ, PIK3IP1, endothelin receptor A, AKT

## Abstract

Oxidative stress, caused by the accumulation of reactive oxygen species (ROS) during acute myocardial infarction (AMI), is one of the main factors leading to myocardial cell damage and programmed cell death. Phosphatidylinositol-3-kinase-AKT (PI3K-AKT) signaling is essential for regulating cell proliferation, differentiation, and apoptosis. Phosphoinositide-3-kinase (PI3K)-interacting protein 1 (PIK3IP1) is an intrinsic inhibitor of PI3K in various tissues, but its functional role during AMI remains unknown. In this study, the anti-ischemic role of PIK3IP1 in an in vitro AMI setting was evaluated using H9c2 cells. The MTT assay demonstrated that cell viability decreased significantly via treatment with H_2_O_2_ (200–500 μM). The TUNEL assay results revealed substantial cellular apoptosis following treatment with 200 μM H_2_O_2_. Under the same conditions, the expression levels of hypoxia-inducible factor (HIF-1α), endothelin-1 (ET-1), bcl-2-like protein 4 (BAX), and cleaved caspase-3 were elevated, whereas those of PIK3IP1, LC3II, p53, and Bcl-2 decreased significantly. PIK3IP1 overexpression inhibited H_2_O_2_-induced and PI3K-mediated apoptosis; however, PIK3IP1 knockdown reversed this effect, suggesting that PIK3IP1 functions as an anti-apoptotic molecule. To identify both the upstream and downstream molecules associated with PIK3IP1, ET-1 receptor type-specific antagonists (BQ-123 and BQ-788) and PI3K subtype-specific antagonists (LY294002 and IPI-549) were used to determine the participating isoforms. Co-immunoprecipitation was performed to identify the binding partners of PIK3IP1. Our results demonstrated that ROS-induced cardiac cell death may occur through the ETA-PI3Kγ-AKT axis, and that PIK3IP1 inhibits binding with both ETA and PI3Kγ. Taken together, these findings reveal that PIK3IP1 plays an anti-ischemic role by reducing the likelihood of programmed cell death via interaction with the ETA-PI3Kr-AKT axis.

## 1. Introduction

Cardiovascular diseases such as acute myocardial infarction (AMI), dilated cardiomyopathy, and congestive heart failure are the leading causes of death worldwide [1]. In AMI, oxidative stress induces the production of large amounts of reactive oxygen species (ROS), impairs the functioning of vascular tissues, and leads to cardiac cell death. However, the major pathways involved in AMI pathogenesis remain unknown. Thus, in the present study, we aimed to investigate the anti-apoptotic pathways involved in AMI.

Phosphoinositide-3-kinase (PI3K) is a crucial hub protein involved in cell survival, growth, apoptosis, proliferation, and immune responses in adaptive and maladaptive cardiac hypertrophy [2]. PI3K converts phosphatidylinositol-4,5-bisphosphate (PIP2) to phosphatidylinositol (3,4,5)-tris-phosphate (PIP3). After phosphorylation, PIP3 recruits and promotes the activation of downstream effectors such as PDK1 (phosphoinositide-dependent kinase-1) and protein kinase B (AKT) [3]. PI3K is a holoenzyme with four distinct subtypes, which include the class 1A subtypes PI3Kα, PI3K_β_, and PI3K_δ_, and the class 1B subtype PI3Kγ [4]. Upon PI3Kα activation in response to the stimulation of receptor tyrosine kinase by agonists such as insulin-like growth factor 1 (IGF-1), AKT induces compensatory physiological hypertrophy [5]. In contrast, PI3Kγ activation negatively regulates cardiac contractility during pressure overload [6]. Moreover, p110γ is activated as a result of G-protein-coupled receptor (GPCR) stimulation by agonists such as endothelin-1 (ET-1) [7], suggesting that PI3Kγ-mediated signaling is associated with pathogenic processes such as cellular apoptosis.

PI3K-interacting protein 1 (PIK3IP1) is a transmembrane protein that negatively regulates PI3K activity, thereby inhibiting AKT phosphorylation [8]. Our previous study investigated the functional role of PIK3IP1 in IGF1-mediated physiological hypertrophy, in which the class 1A PI3Kα isoform is involved [9]. However, the functional role of PIK3IP1 in GPCR-mediated pathological hypertrophy, in which the class 1B isoform PI3Kγ is involved, remains unknown. Our preliminary study using a PIK3IP1-deficient mouse model suggests that PIK3IP1 plays an anti-ischemic role (Appendix A).

ET-1 activates several signal transduction pathways linked to cellular hypertrophy, growth, migration, and proliferation in several cell types, including cardiac tissues [10]. In cardiac cells, ET-1 may play an important role in the pathophysiology of chronic heart failure and ischemic heart diseases [10]. Using a rat model of myocardial infarction (MI), such as myocardial stunning, Klainguti et al. (2000) reported that a surge in the myocardial ET-1 level after 20 min of ischemia led to apoptosis in the post-ischemic condition [11]. Additionally, during hypoxia, the expression of ET-1 is elevated by hypoxia-inducible factor 1α (HIF-1α) [12]. ET-1 exerts its biological action through the activation of two receptor subtypes—endothelin A receptor (ET_A_) and endothelin B receptor (ET_B_), both of which belong to a large family of GPCRs. The activation of each receptor subtype, however, exerts opposing pathophysiological effects. For instance, the ET_A_ receptor leads to vasoconstriction and exerts pro-oxidative actions, whilst the stimulation of ET_B_ may elicit vasodilatation in healthy subjects due to the stimulation of nitric oxide production [13].

The present study used H9c2 cells and H_2_O_2_ treatment to examine the pathways underlying ROS-induced cardiac programmed cell death, focusing on the anti-ischemic effects of PIK3IP1. Our results suggest that PIK3IP1 plays an essential protective role in ischemic hypoxia-induced cardiac cell apoptosis via interaction with ET_A_-PI3Kγ/AKT-mediated signaling.

## 2. Materials and Methods

### 2.1. Chemicals and Antibodies

BQ-123, BQ-788, IPI-549, and LY294002 were purchased from Sigma-Aldrich (St. Louis, MO, USA). H_2_O_2_ was dissolved in distilled water and BQ-123, BQ-788, LY294002, or IPI-549 was dissolved in 0.01% DMSO to make the stock solutions. The antibodies used are described in Appendix A.

### 2.2. Cell Culture and Treatment

H9c2 rat cardiomyocyte-derived cells were obtained from American Type Culture Collection (ATCC, Manassa, VA, USA). H9c2 rat cardiomyocyte-derived cells were placed in a serum-free medium without antibiotics for 24 h prior to siRNA transfection. Cells were then transfected with 25 nM siRNA for PIK3IP1 and negative control (Bioneer, Daejeon, Korea) using DharmaFECT-1 reagent (Dharmacon, Lafayette, CO, USA) according to the manufacturer’s instructions. After 48 h, cells were treated with H_2_O_2_ (200 nM) or with blank for 3 h or 24 h, respectively. PIK3IP1-overexpressing clone was purchased from GenScript (Piscataway, NJ, USA), and was constructed using the pcDAN3.1+/C-(K)-DYK vector. H9c2 cells were transfected with PIK3IP1-overexpressing (PIK) and pcDNA3.1 control (Vec) using DharmaFECT-1 reagent (Dharmacon) according to the manufacturer’s instructions.

### 2.3. MTT Assay

Cell viability was determined using the MTT assay, which is based on the enzymatic reduction of 3-[4,5-dimethylthiazole-2-yl]-2,5-diphenyltetrazolium bromide (MTT) to MTT-formazan by mitochondrial dehydrogenases in viable cells. H9c2 cells were seeded into a 96-well plate at a density of 1 × 10^5^ cells per well and treated with H_2_O_2_ or DMSO. After incubating for 24 h, 50 μL of MTT stock solution (2 mg/mL) was added to each well to yield a total reaction volume of 250 μL. After incubating for a further 4 h, the supernatants were aspirated, and the formazan crystals in each well were dissolved in dimethyl sulfoxide (150 μL). The absorbance at 540 nm was read on a spectrophotometer.

### 2.4. Western Blot Analysis

Total cell lysates were prepared by scraping the cells in 200 µL 1× RIPA lysis buffer (Invitrogen, Eugene, OR, USA). Proteins separated on the gels were transferred to polyvinylidene fluoride membranes. The membranes were blocked using 5% bovine serum albumin (BSA; Sigma-Aldrich) in TBST for 1 h at room temperature. Blots were cut prior to hybridization with antibodies and incubated at 4 °C overnight with primary antibodies. The membranes were developed using ECL Advance Western Blotting Detection Kit (GE Healthcare, Little Chalfont, Buckinghamshire, UK) using an ImageQuant LAS 4000 mini (GE Healthcare). Quantitation was performed using ImageJ software (NIH Image). Uncropped gel images are shown in Appendix A.

### 2.5. TUNEL Assay

DNA fragmentation was detected in situ using a TUNEL assay kit (In Situ Cell Death Detection Kit, TMR red; Roche Applied Science, Penzberg, Germany), as described previously [9]. Briefly, Cells were fixed with 4% paraformaldehyde in PBS for 1 h at room temperature, and permeabilized with 0.2% triton X-100 for 2 min on ice. Nuclear staining was performed using Hoechst (Molecular Probes, Eugene, OR, USA).

### 2.6. Co-Immunoprecipitation

The cells were lysed in 400 µL of NP-40 buffer with protease inhibitors and immunoprecipitated with Dynabeads protein G (Invitrogen, 10003D). The experiment was continued according to the manufacturer’s instructions, beginning with the addition of 10 μL of antibodies or 10 μL of rabbit IgG to create the Co-IP bead complex, by using a magnetic rack, and added to the lysate. The bound proteins were eluted and the samples were subjected to Western blotting.

### 2.7. Quantitative Real-Time PCR (qRT-PCR)

TRI reagent (Sigma-Aldrich) was used for total RNA extraction. Reverse-transcriptase reactions were performed using PrimeScript RT Master Mix (Takara Bio, Otsu, Japan) with oligo-dT priming. qRT-PCR was performed using a Step One Plus real time PCR system (Applied Biosystems, Foster City, CA, USA) with SYBR Green (Kapa Biosystems, Boston, MA, USA) as a fluorescent dye. The primer sequences are shown in Appendix A.

### 2.8. Statistics

Statistical analyses were performed using Prism 8 software (GraphPad software, SanDiego, CA, USA). All data are shown as mean ± standard deviation (SD) for each group. For statistical comparison of the two groups, an unpaired Student’s t-test was used. For multiple comparisons, one-way analysis of variance (ANOVA) with Bonferroni correction was used. All experiments were performed at least three times. A value of *p* < 0.05 was considered statistically significant.

## 3. Results

### 3.1. H_2_O_2_-Induced Apoptosis of H9c2 Cardiomyocytes Is Accompanied by Decreased PIK3IP1 and Bcl-2 Expression, but Increased HIF-1α, ET-1, p-AKT, BAX, and Cleaved Caspase-3 Expression

H9c2 cardiomyocytes were treated with the indicated concentrations (10–500 μM) of H_2_O_2_ for 24 h, and the cell viability was examined using the MTT assay to determine the optimal concentration of H_2_O_2_ for the subsequent experiments (Figure 1A). Cell viability decreased in a dose-dependent manner, with 200 and 500 μM H_2_O_2_ substantially reducing cell viability. Therefore, 200 μM H_2_O_2_ was selected for use in the TUNEL assay to measure apoptosis-induced DNA fragmentation as an inducer of apoptosis. The results demonstrated that the number of TUNEL-positive cells increased substantially after 24 h treatment with 200 μM H_2_O_2_ (Figure 1B). Immunoblotting was performed to investigate the effects of 200 μM H_2_O_2_ treatment on the expression of proteins involved in the ROS-mediated apoptosis pathways in cardiac cells (Figure 1C). The transcription factor HIF-1α plays an important role in cellular responses to systemic oxygen levels in mammals [14]. Our results showed that HIF-1α expression increased within 1 h in response to 200 μM H_2_O_2_ treatment; however, this effect was partially reversed with prolonged incubation (3–24 h). Interestingly, the expression of ET-1 gradually increased in the presence of 200 μM H_2_O_2_, but that of PIK31P1 decreased within the same time range. The protein level of p-AKT showed a bell-shaped time-dependent curve. The expression levels of p53 and LC3-II, the autophagosomal markers, showed time-dependent increases. The levels of pro-apoptotic proteins, BAX and cleaved caspase-3 (c-Casp3), increased in a similar manner to those of ET-1; consistently, the levels of antiapoptotic protein Bcl-2 decreased. These results suggest that ET-1 acts as a pro-apoptotic protein through PI3Kγ-AKT and pathogenic autophagy and apoptosis signaling [15], whereas PIK3IP1 may act as an anti-apoptotic protein under ROS-rich ischemic conditions.

### 3.2. PIK3IP1 Overexpression Inhibits H_2_O_2_-Induced Cell Death via Downregulation of p110α, p110γ, p-AKT, and Cleaved Caspase-3 Expression

To confirm the anti-apoptotic role of PIK3IP1 in ROS-generating conditions, PIK3IP1 was overexpressed as described in Materials and Methods. Western blot analysis revealed a specific band (DYKDDD) for PIK3IP1 in pcDNA3.1-PIK3IP1-transfected (PIK) cells, but not in the pcDNA3.1-transfected control (Vec) cells (Figure 2A). The total expression of PIK3IP1 in PIK cells was approximately two times higher than that of Vec cells. H_2_O_2_ treatment decreased the expression of PIK3IP1 in both the PIK and Vec cells. H_2_O_2_-induced upregulation of p110α, p110γ, p-AKT, p53, and LC3-II expression was significantly decreased in PIK3IP1-overexpressing cells (Figure 2A). The level of cleaved caspase-3 protein, a cell death marker, was also significantly decreased in PIK3IP1-overexpressing cells, suggesting that PIK3IP1 overexpression could inhibit cardiomyocyte cell death. The number of TUNEL-positive cells was significantly increased in the H_2_O_2_-treated cells, but decreased significantly in the PIK3IP1-overexpressing cells (Figure 2B). These results suggest that PIK3IP1 plays an anti-apoptotic role by inhibiting PI3K-AKT signaling during H_2_O_2_ exposure.

### 3.3. PIK3IP1 Knockdown Increased H_2_O_2_-Induced Cell Death via Upregulation of p110α, p110γ, p-AKT, and Cleaved Caspase-3 Expression

To determine whether the downregulation of PIK3IP1 levels would exacerbate cardiomyocyte cell death via inhibition of PI3K, PIK3IP1 knockdown (KD) was performed using siRNA. The expression of PIK3IP1 was significantly decreased in siPIK-transfected cells compared to siNC-transfected cells (by approximately 50%) (Figure 3A). The expression of p110α, p110γ, p-AKT, p53, LC3-II, and cleaved caspase-3 proteins increased significantly in siPIK-transfected cells. The addition of 200 μM H_2_O_2_ showed additive effects on the protein levels (Figure 3A), suggesting that H_2_O_2_ increases cardiomyocyte cell death in PIK3IP1-deficient cells by directly activating the PI3K-AKT pathway. The number of TUNEL-positive cells significantly increased in the H_2_O_2_-treated group, which was amplified in the PIK3IP1 KD group (Figure 3B). This indicates that the inhibition of PIK3IP1 induced apoptotic signaling and aggravated H_2_O_2_-induced apoptosis.

### 3.4. H_2_O_2_ Induced Cardiomyocyte Cell Death via ET_A_-PIK3IP1 Binding

Increasing evidence suggests that ET-1 is involved in the processes of myocardial ischemia and infarction. For instance, a 2–10-fold increase in ET-1 concentration has been reported during the early onset of myocardial infarction [15,16]. Under hypoxic conditions, HIF-1α expression activates ET-1 gene expression, thereby increasing ET-1 levels; however, the effects of ET-1 on apoptosis in cultured cardiomyocytes remains controversial [17]. To investigate the relationships between ET_A_, PIK3IP1, and p110γ, co-immunoprecipitation experiments were conducted. Figure 4A,B show that both p110γ and p110α were co-immunoprecipitated with PIK3IP1; however, ET_A_, but not ET_B_, was co-immunoprecipitated with PIK3IP1 (Figure 4C,D).

The number of TUNEL-positive cells significantly increased in the H_2_O_2_-treated group; this effect was inhibited in the BQ-123 (ET_A_ inhibitor)-treated group, but not in BQ-788 (ET_B_ inhibitor)-treated cells (Figure 5A). These results imply that ET_A_ receptors, but not ET_B_ receptors, mediate hypoxia-induced injury through endogenous ET-1. Furthermore, following PIK3IP1 KD, 3 h incubation with BQ-123 rescued the protein expression of PIK3IP1 (Figure 5B), suggesting that the ET_A_ inhibitor decreased the efficiency of siRNA interference.

### 3.5. H_2_O_2_ Induces H9c2 Cell Death through the ET_A_-PIK3IP1-PI3Kγ Axis

To verify the preferential role played by ET_A_ in H_2_O_2_-induced cardiac cell apoptosis, the downstream PI3K-AKT signaling molecules were examined in the presence or in the absence of the ET_A_- and ET_B_-specific inhibitors (BQ-123 and BQ-788, respectively). The expression of p110α and p110γ isoforms was upregulated following H_2_O_2_ treatment, however their expression levels diminished in the BQ-123-pretreated group. BQ-788 pretreatment decreased p110α expression, but not that of p110γ (Figure 6A). These data suggest that p110γ is sensitive only to ET_A_, but not to ET_B_. Further experiments were conducted using LY294002, a p110α-specific inhibitor, and IPI-549, a p110γ-specific inhibitor. The downstream p-AKT and cleaved caspase-3 levels decreased in the LY294002- and IPI-549-treated groups. However, the degree of inhibition was greater in the IPI-549-treated p110γ group than in the LY294002-treated p110α group (Figure 6B). These findings, altogether, imply that the PIK3IP1-PI3Kγ-AKT axis plays an essential role in H_2_O_2_-induced cardiomyocyte cell death.

## 4. Discussion

Acute myocardial infarction (AMI) is a serious health problem worldwide with a high mortality rate. In AMI, the myocardial ischemia produces a large amount of ROS due to oxidative stress, causing serious damage to the myocardial cell membrane and apoptosis of cardiomyocytes [18]. The suppression of apoptosis is important for reducing myocardial infarct size. However, the essential signaling pathways associated with AMI-induced apoptosis remain to be studied. The present study focused on identifying the molecular players, and their interaction cascades, involved in the ROS-induced apoptosis pathways using the H9c2 cell line.

Most studies concerning the apoptotic pathways in the heart have come from cultured cells, such as primary neonatal rat cardiomyocyte, the atrial cardiac muscle derived HL-1 cell line, and the cardiomyocyte-like myoblast cell line H9c2 [19]. Since H9c2 cells are known to be more sensitive to hypoxia than HL-1 cells [20], the present study utilized a well-characterized H9c2 cell line derived from rat cardiac myoblasts. The results displayed in Figure 1 indicate that H9c2 cell viability substantially decreased following treatment with 100–500 µM H_2_O_2_, which was associated with elevated programmed cell death. Western blot analysis showed that HIF-1α expression was upregulated within 1 h of treatment, followed by the upregulation of ET-1, p-AKT, and mitochondrial apoptosis markers such as BAX and cleaved Caspase-3. In contrast, the expression of anti-apoptotic PIK3IP1 and Bcl-2 proteins was downregulated, suggesting that the in vitro experimental system with H9c2 cells was well established for the present study.

Autophagy is a well-known protective mechanism generally, but the uncontrolled activation of autophagy can be detrimental [21]. The mouse study of ischemia/reperfusion (I/R) showed evidence of activated autophagy via the accumulation of autophagosomes and autolysosomes, suggesting the contribution of autophagic cell death to the I/R injury. p53-induced upregulation of autophagic cell death could be associated with the BCL2 Interacting Protein 3 (Bnip3) [22]. The data showing the H_2_O_2 –_induced upregulation of p53 and LC3-II expression in Figure 1 suggest that H_2_O_2_ could induce autophagic cell death in hypoxic conditions.

PI3K signaling plays an important role in the pathogenesis of cardiovascular diseases [23,24]. However, different types of PI3K isoforms may perform distinct functions. For example, PI3Kα may play an important role in the induction of physiological cardiac hypertrophy upon exercise [25], whereas PI3Kγ may play a pathological role in the induction of congestive heart failure, considering the PI3Kγ-mediated inhibition of cardiac contractility in failing hearts [26,27].

Previous studies demonstrated that PI3Kγ^−/−^ mice displayed increased ischemia-reperfusion injury, and PI3Kγ kinase-dead knock-in (PI3Kγ^KD/KD^) mice exhibit a cardiac injury similar to wild-type animals [28]. This suggests a protective role of PI3Kγ in myocardial ischemia–reperfusion injury in mice, mediated via a kinase-independent mechanism. Figure 6B, however, shows that PI3Kγ blockade via the use of IPI-549, whereby PI3Kα/AKT was only fully activated by H_2_O_2_, reduced the degree of apoptosis considerably, suggesting that PI3Kγ, but not PI3Kα, plays the major role in ROS-induced cell death. Notably, our group also found, first, that PI3Kγ interacts directly with PIK3IP1 to regulate programmed cell death (Figure 4B). Although p-Akt is a known survival kinase generally, it may also facilitate apoptosis under different conditions in various tissues [29,30].

The underlying mechanisms for the activation of the pathologic conditions by PI3Kγ are not fully elucidated. However, it appears that the activation of the pathologic gamma subunit of PI3K could activate the p53-dependent sequential activation of autophagy and apoptosis in the same cells (Figure 1). It is also important to note that we increased the concentration of H_2_O_2_ to a certain degree to induce apoptosis, to mimic the whole animal (mouse) myocardial infarction model, where the increased apoptosis and increased expression of tissue cleaved caspase-3 occurred (the data are not shown).

Akt, the downstream effector of PI3K is an important activator of eNOS [31]. eNOS-derived NO exerts vasodilatory, anti-inflammatory, and anti-proliferative effects via cGMP-dependent protein kinase (PKG) by increasing the cGMP levels in biological systems [32]. Among the various downstream effectors of PKG, VASP regulation in cancer or cardiovascular diseases has been studied due to its role in cell adhesion, migration, and proliferation [33,34]. According to a recent paper, NO-induced VASP phosphorylation at serine 239 by PKG is required for the anti-proliferative effect of NO [35]. Appendix A shows that H_2_O_2_-induced p-VASP may be involved in the growth-inhibitory effects of NO on H9c2. These findings appear to be implicated in the H_2_O_2_-associated upregulation of the eNOS-PKG-VASP pathway. Furthermore, the downregulated SOD1, SOD2, GPX1, and GPX4 mRNA levels by H_2_O_2_ were recovered by PIK3IP1 overexpression, which suggests that PIK3IP1 is an anti-ischemic protein (Appendix A).

PIK3IP1 was recently identified as a transmembrane protein that shares its homology with the PI3K regulatory subunit p85 [36]. This protein directly interacts with p110, the catalytic subunits of PI3K, and regulates PI3K activity through its p85-like domain. Previous studies showed that PIK3IP1 could bind to class IA PI3K subtypes such as PI3Kα and PI3K_β_ [36]. Our data now demonstrate that class IB PI3K subtype PI3Kγ can also interact with PIK3IP1 (Figure 4B), suggesting that class IB regulatory subunit p101 or p84 could also share their homology with PIK3IP1.

PIK3IP1 is abundantly expressed in many tissues, including the heart, liver, brain, and lungs. Overexpression of PIK3IP1 in hepatocytes leads to the inhibition of PI3K signaling and the suppression of hepatocyte carcinoma development [8]. PIK3IP1 is involved in the PI3K pathway, which is associated with many cellular functions, such as T cell activation, carcinogenesis, and apoptosis [37]. DeFrances et al. (2012) reported that silencing PIK3IP1 increases PI3K activity in basal conditions [38]. We previously reported the function of PIK3IP1 in physiological cardiac hypertrophy [9], but the function of PIK3IP1 in the heart under pathological conditions remained unknown. In the present study, we found that PIK3IP1 overexpression inhibited cardiomyocyte cell death via downregulation of PI3K-AKT signaling (Figure 2), while PIK3IP1 KD exhibited increased cardiomyocyte cell death via upregulation of the signaling pathway (Figure 3). According to our unpublished data, using the regression model [39,40], the expression level of PIK3IP1 was markedly downregulated by trans-aortic constriction (TAC), but rapidly increased after the release of TAC. This finding suggests that PIK3IP1 expression is dynamically changed by blood pressure. Interestingly, PIK3IP1 expression was also decreased upon initiation of myocardial infarction, but was reversed with increased duration (unpublished data), further supporting the idea that PIK3IP1 is closely related to the pathophysiological condition of the heart. Indeed, in myocardial infarction (MI) using conditionally PIK3IP1-deficient mice, PIK3IP1 has an anti-ischemic effect, as shown in Appendix A.

The use of an ET_A_ receptor antagonist exerts beneficial effects on chronic heart failure, as evidenced by a reduction in infarct size, improved reperfusion, coronary flow, and protection during ischemic-reperfusion injury [41,42,43]. The results of the studies blocking ET_B_ receptors have been inconsistent. Our data, showing the inhibition of ROS-induced cardiac apoptosis by an ET_A_ antagonist (BQ-123), but not by an ET_B_ antagonist (BQ-788) (Figure 5A), suggest that ET_A_ is important for the signaling cascade, but not ET_B_. Our Co-IP data also showed that ET_A,_ but not ET_B_, could interact with PIK3IP1 (Figure 4C, D). However, further studies will be required to clarify the molecular basis of the interactions. Furthermore, the results displayed in Figure 6C indicate that PI3Kγ, but not PI3Kα, can mainly modulate ET_A_ in the signaling cascade.

Major treatments for various heart diseases include traditional pharmacotherapy and open heart surgeries that can cause inevitable side-effects. Recent innovations in gene editing and cell therapy technologies have shown promising trends of treatment focusing on specific genes [44,45]. For instance, recent studies have shown that FOXO3 genetically engineered human mesenchymal progenitor cells confer better therapeutic efficacy in a mouse model of myocardial infarction [46,47]. Our present study of PIK3IP1 that shows the functional anti-ischemic role in the heart will be immensely beneficial to gene therapy efforts for ischemic heart diseases.

In conclusion, the present study provides evidence to suggest that ROS-induced cardiac cell death is mediated by the ET_A_-PIK3IP1-PI3Kγ axis. Under normoxic conditions, PIK3IP1 expression is relatively high. This is due to the lower inhibition by ET_A_; hence, subdued PI3Kγ activity is retained because of the intense inhibition by PIK3IP1. On the other hand, under hypoxic conditions, when ROS is generated from mitochondria, the signaling direction is reversed, and the likelihood of programmed cell death increases. Thus, the present study provides evidence that PIK3IP1 plays an essential role in regulating pathogenic signals in the heart. Hence, it is possible that PIK3IP1 will be the subject of gene therapy in the future.

## Figures and Tables

**Figure 1 cells-11-02162-f001:**
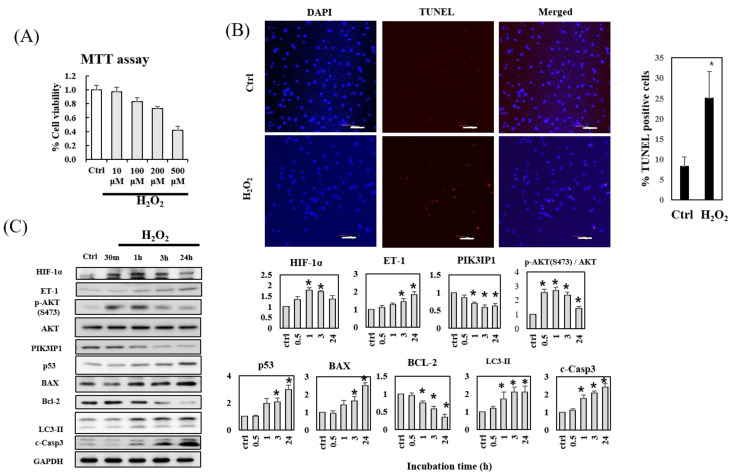
H_2_O_2_-induced apoptosis of H9c2 cardiomyocytes is accompanied by decreased PIK3IP1 and Bcl2 expression, but increased HIF-1α, ET-1, p53, BAX, LC3-II, and cleaved caspase-3 expression. (**A**) H9c2 cardiomyocytes were treated with the indicated concentrations of H_2_O_2_ for 24 h. Cell viability was measured using the MTT assay. (**B**) DNA fragmentation as an indication of programmed cell death was detected via TUNEL assay, and cells were counterstained with DAPI for nuclei staining. The scale bar represents 200 μm. The graph on the right shows the significantly increased TUNEL-positive cells following H_2_O_2_ treatment. (**C**) The protein expression levels in the control (ctrl) and in the H_2_O_2_-treated samples were measured via Western blotting. The gel band intensities were normalized using GAPDH for quantifications and are shown on the right. The results are presented as means ± SEM; n = 3–5; statistical significance is shown as * *p* < 0.05 relative to control groups (ctrl).

**Figure 2 cells-11-02162-f002:**
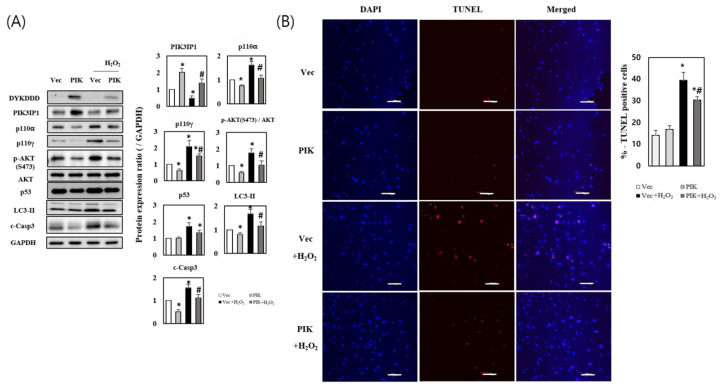
PIK3IP1 overexpression inhibits H_2_O_2_-induced cell death via downregulation of p110α, p110γ, p-AKT, p53, LC3-II, and cleaved caspase-3 expression. H9c2 cardiomyocytes were transfected with pcDNA3.1 (Vec) or pcDNA3.1-PIK3IP1 (PIK) for 24 h, and subsequently treated with or without H_2_O_2_ (200 μM). (**A**) The transfected vector expression was detected using anti-DYKDDDDK tag antibody. The protein levels were measured via Western blotting. (**B**) The DNA fragmentation as an indication of programmed cell death were detected via TUNEL assay and the cells were counterstained with DAPI (nuclei staining). The scale bar represents 200 μm. The results for data displayed in bar charts are presented as means ± SEM; n = 3–5; statistical significance is shown as * *p* < 0.05 relative to Vec group. ^#^ *p* < 0.05 relative to Vec treated with H_2_O_2_.

**Figure 3 cells-11-02162-f003:**
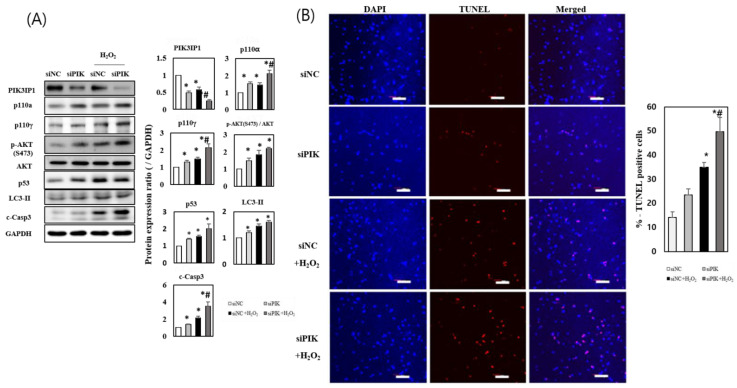
PIK3IP1 knockdown increased H_2_O_2_-induced cell death via upregulation of p110α, p110γ, p-AKT, p53, LC3-II, and cleaved caspase-3 expression. H9c2 cells were transfected with siPIK (si-PIK3IP1) or siNC (negative control) for 24 h (25 nM each), and subsequently treated with or without H_2_O_2_ (200 μM). (**A**) The protein levels were measured via Western blotting. (**B**) The DNA fragmentation as an indication of programmed cell death was detected via TUNEL assay, and the cells were counterstained with DAPI (nuclei staining). The scale bar represents 200 μm. The combined results displayed in bar charts are presented as mean ± SEM; n = 3–5; statistical significance is shown as * *p* < 0.05 relative to Vec group. ^#^ *p* < 0.05 relative to Vec treated with H_2_O_2_.

**Figure 4 cells-11-02162-f004:**
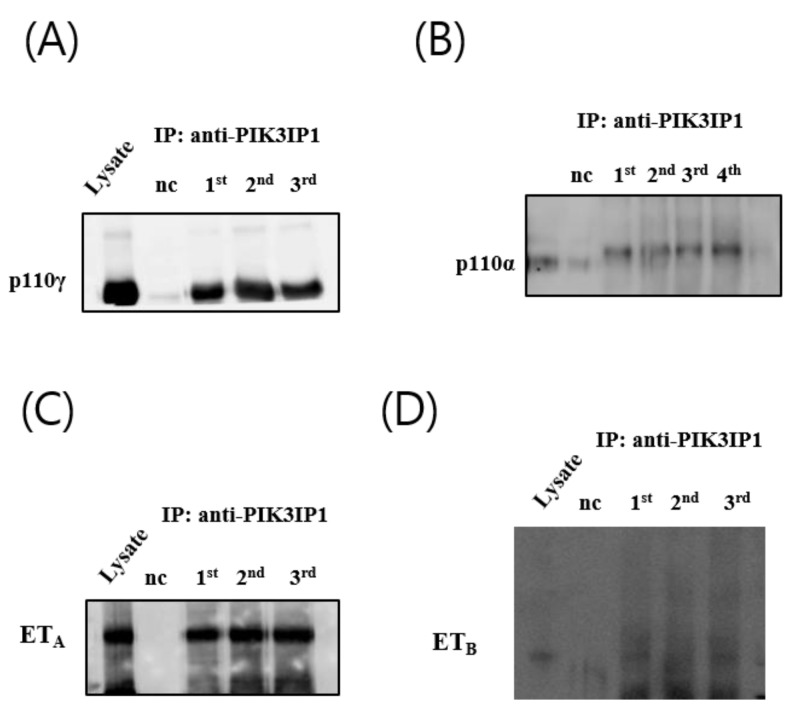
PIK3IP1 binds with p110α, p110γ, and ET_A_, but not with ET_B_. Co-immunoprecipitation (Co-IP) assays using H9c2 cells were detected via Western blotting using the anti-PIK3IP1 and (**A**) anti-p110α antibody, (**B**) anti-p110γ antibody, (**C**) anti-ET_A_ antibody, and (**D**) anti-ET_B_ antibody. The mock IP negative control experiment was performed by incubating cell extracts with G Dynabeads coupled with rabbit IgG.

**Figure 5 cells-11-02162-f005:**
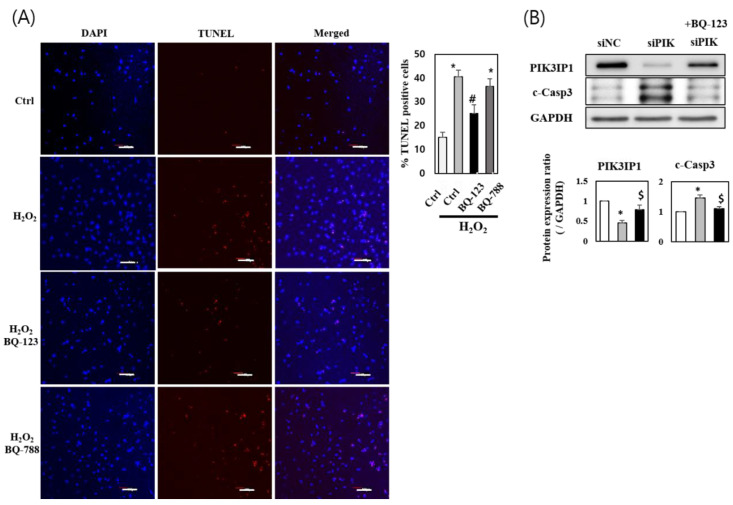
H_2_O_2_ induced cardiomyocyte cell death via activation of ET_A_-PIK3IP1 binding: (**A**) H9c2 cells were treated with H_2_O_2_ (200 μM) for 24 h after pretreatment with BQ-123 (10 μM) or BQ-788 (10 μM) or DMSO. Fragmentation was detected via TUNEL assay and cells were counterstained with DAPI for nuclei staining. The scale bar represents 200 μm. (**B**) H9c2 cells were transfected with siPIK (si-PIK3IP1) or siNC (negative control) for 24 h (25 nM each), and subsequently treated with or without BQ-123 (10 μM) for 3 h. The protein expression levels were measured via Western blotting. Results are presented as mean ± SEM; n = 3–5; statistical significance is shown as * *p* < 0.05 relative to control group (Ctrl or siNC). ^#^ *p* < 0.05 relative to H_2_O_2-_treated group. ^$^ *p* < 0.05 relative to siPIK group.

**Figure 6 cells-11-02162-f006:**
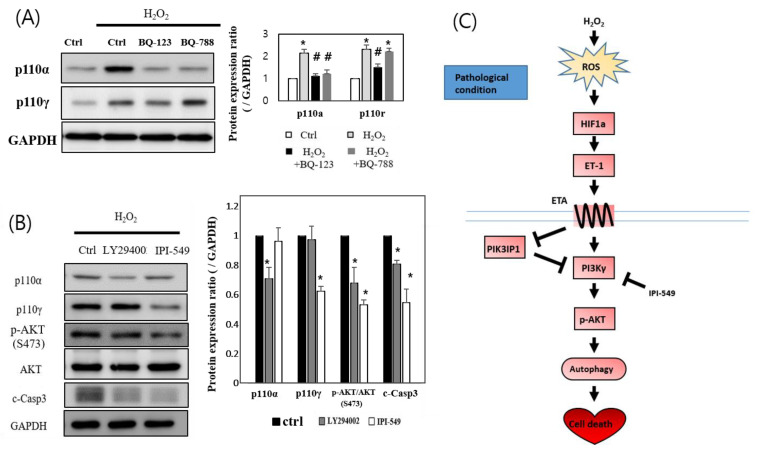
H_2_O_2_ induces H9c2 cell death through the ET_A_-PIK3IP1-PI3Kγ axis. (**A**) H9c2 cells were treated with H_2_O_2_ (200 μM) for 3 h after pretreatment with BQ-123 (10 μM) or BQ-788 (10 μM) or DMSO. The protein expression levels were measured via Western blotting. (**B**) Cells were treated with H_2_O_2_ for 3 h after pretreatment with 20 μM LY294002 (LY) or 10 μM of IPI-549 (IPI). The expression levels were measured via Western blotting. Results are presented as mean ± SEM; n = 3–5; statistical significance is shown as * *p* < 0.05 relative to control group. ^#^ *p* < 0.05 relative to H_2_O_2-_treated group. (**C**) The proposed model shows that the anti-ischemic effects of PIK3IP1 in reducing the likelihood of cardiac cell death are mediated through its interactions with the ETA-PI3Kγ-AKT axis.

## Data Availability

Not applicable.

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
