# Peer review of "Anti-Ischemic Effects of PIK3IP1 Are Mediated through Its Interactions with the ETA-PI3Kγ-AKT Axis"

_cells, 2022, doi:10.3390/cells11142162_

Round 1
Reviewer 1 Report
In this manuscript “Anti-ischemic effects of PIK3IP1 are mediated through its interactions with the ETA-PI3Kγ-AKT axis”, the authors used H9c2 cells and hydrogen peroxide treatment to mimic the oxidative stress generated during acute myocardial infarction. They found that PIK3IP1 plays an essential protective role in ischemic hypoxia-induced cell apoptosis through interacting with ETA-PI3Kr-AKT mediated pathway. In general, the manuscript is well designed and organized. However, there are some issues need to be solved:
Major comment:
1. Line 146, based on the first paragraph of Result section and Figure 1, the authors can’t draw the conclusion that PIK3IP1 serves as an anti-apoptotic protein during ROS-rich ischemic condition. This conclusion should be toned down.
2. To further highlight the significance of this paper, the authors need to test the protective effect of BQ-123 treatment in acute myocardial infarction mouse/rat model.
Minor comment:
1. In Method section, lack of MTT assay description.
2. Figure 1A, lack of statistical analysis.
3. The quality of immunofluorescence images in this paper should be improved.
4. Figure 2A, the bands of P53, AKT and GAPDH are overexposed.
5. There is a legend of Figure 3C, but I can’t find the corresponding figure.
6. In Figure 4B, lack of input/cell lysate sample. If left-most lane is Input sample, why the protein sizes are different between input and IP group?
7. In the discussion section (line 349), the authors mentioned that they display their results in Graphic abstract, but I can’t find this graph.
8. In the discussion section, the authors could discuss the research progress about the therapeutic strategy about ischemia (for example PMID: 32809106 and 30661960) and compare the feasibility with their findings.
Reviewer 2 Report
The article entitled “Anti-ischemic effects of PIK3IP1 are mediated through its inter-2 actions with the ETA-PI3Kγ-AKT axis” has several problems, starting from the title. Indeed, there are no ischemia-studies in the manuscript and the acronyms need to be explained.
The authors study the H2O2 cardiac-injury and role of ETA-PI3Kγ-AKT axis. The authors may know that oxidants and antioxidants have dual roles in cardio-protection and cardiac injury. It is well known that cardioprotective strategies able to attenuate myocardial I/R injury (e.g. preconditioning (IP) and postconditioning (PostC)), can not only reduce the size of myocardial infarction and arrhythmia, but also improve the contractile function of myocardium after ischemia-reperfusion [Circulation, vol. 74, no. 5, pp. 1124-1136, 1986; Antioxid Redox Signal, vol. 14, no. 5, pp. 833-850, 2011; Am J Physiol Heart Circ Physiol, vol. 285, no. 2, pp. H579-588, 2003.]. Meanwhile, some substances can be used as an adjuvant therapy to mitigate myocardial I/R injury through their antioxidant properties [Pharmacol Res, vol. 61, no. 4, pp. 342-348, 2010; Pharm Biol, vol. 55, no. 1, pp. 1144-1148, 2017; Sci Rep, vol. 6, pp. 36435, 2016; Mol Cell Biochem, vol. 474, no. 1-2, pp. 171-180, 2020]. However, if antioxidants are used indiscriminately to quench reactive species (ROS/RNS) in I/R injury, it may interfere with normal physiological redox signals, resulting in damage to cell functions [Basic Res Cardiol, vol. 108, no. 6, pp. 392, 2013.]. For instance, some studies showed that in the trigger stage of IP and PostC, the production of ROS/RNS was beneficial to activate PI3K and PKC to form memory effects and induce the pathway of RISK (Reperfusion Injury Salvage Kinase) and SAFE (Survival Activating Factor Enhancement) to inhibit the opening of mPTP during reperfusion [Am J Physiol Heart Circ Physiol, vol. 295, no. 2, pp. H874-882, 2008. Br J Pharmacol, vol. 172, no. 8, pp. 1974-1995, 2015; Basic Res Cardiol, vol. 104, no. 2, pp. 189-202, 2009]. Obviously, some antioxidants, given in this phase, will block IP and PostC cardio-protection [Am J Physiol Heart Circ Physiol, vol. 284, no. 2, pp. H698-703, 2003; Circ Res, vol. 88, no. 8, pp. 802-809, 2001. Basic Res Cardiol, vol. 108, no. 6, pp. 392, 2013]. Therefore, according to the active phase of ROS/RNS, the appropriate time of oxidants and antioxidants should be determined after taking into full consideration of their dual role in cardio-protection and cardiac injury. In addition, myocardial I/R injury will be aggravated (increasing the level of ROS/RNS and reducing the level of antioxidant enzymes) in the presence of comorbidities such as metabolic syndrome, diabetes, and myocardial hypertrophy [Pharmacol Rev, vol. 59, no. 4, pp. 418-458, 2007]. Therefore, it should be considered to increase the dose of antioxidants to achieve a better therapeutic effect. In conclusion, when oxidants/antioxidants are given it is essential to find the best dose and time of administration. A dose response is required in the present study.
Furthermore (NO-redox interactions), some studies have found that the cardioprotective effects are activated through post translational modification of proteins (S-Nitrosylation, SNO) within the RISK and SAFE pathways [Exp Biol Med (Maywood), vol. 239, no. 6, pp. 647-662, 2014]. Endothelin may interfere with nitric oxide production. This implies that redox state and endothelins may affect SNO. Moreover, the activity of antioxidant enzymes (GSH and SOD) can be influenced by antioxidants/oxidants to exert cardioprotective effects. At the same time, these antioxidant enzymes can limit the amount of protein SNO by denitrosylation, preventing excessive SNO from producing cytotoxicity in a pathological environment of nitrosative stress [Exp Biol Med (Maywood), vol. 239, no. 6, pp. 647-662, 2014; Biochem Biophys Res Commun, vol. 228, no. 1, pp. 88-93, 1996; Arch Biochem Biophys, vol. 361, no. 2, pp. 323-330, 1999; Proc Natl Acad Sci U S A, vol. 109, no. 11, pp. 4314-4319, 2012.]. Consequently, H2O2 may promote SNO and/or inhibit the production of excess SNO to alleviate myocardial I/R injury. And further studies are needed to explore the effect of ETA-PI3Kγ-AKT axis on SNO.
Authors should stress the novelties of their work.
